# Suicide and Associations with Air Pollution and Ambient Temperature: A Systematic Review and Meta-Analysis

**DOI:** 10.3390/ijerph18147699

**Published:** 2021-07-20

**Authors:** Seulkee Heo, Whanhee Lee, Michelle L. Bell

**Affiliations:** School of the Environment, Yale University, New Haven, CT 06511, USA; whanhee.lee@yale.edu (W.L.); michelle.bell@yale.edu (M.L.B.)

**Keywords:** suicide, air pollution, temperature, climate change, mortality, time-series, case-crossover

## Abstract

Given health threats of climate change, a comprehensive review of the impacts of ambient temperature and ar pollution on suicide is needed. We performed systematic literature review and meta-analysis of suicide risks associated with short-term exposure to ambient temperature and air pollution. Pubmed, Scopus, and Web of Science were searched for English-language publications using relevant keywords. Observational studies assessing risks of daily suicide and suicide attempts associated with temperature, particulate matter with aerodynamic diameter ≤10 μm (PM_10_) and ≤2.5 mm (PM_2.5_), ozone (O_3_), sulfur dioxide (SO_2_), nitrogen dioxide (NO_2_), and carbon monoxide (CO) were included. Data extraction was independently performed in duplicate. Random-effect meta-analysis was applied to pool risk ratios (RRs) for increases in daily suicide per interquartile range (IQR) increase in exposure. Meta-regression analysis was applied to examine effect modification by income level based on gross national income (GNI) per capita, national suicide rates, and average level of exposure factors. In total 2274 articles were screened, with 18 studies meeting inclusion criteria for air pollution and 32 studies for temperature. RRs of suicide per 7.1 °C temperature was 1.09 (95% CI: 1.06, 1.13). RRs of suicide per IQR increase in PM_2.5_, PM_10_, and NO_2_ were 1.02 (95% CI: 1.00, 1.05), 1.01 (95% CI: 1.00, 1.03), and 1.03 (95% CI: 1.00, 1.07). O_3_, SO_2_, and CO were not associated with suicide. RR of suicide was significantly higher in higher-income than lower-income countries (1.09, 95% CI: 1.07, 1.11 and 1.20, 95% CI: 1.14, 1.26 per 7.1 °C increased temperature, respectively). Suicide risks associated with air pollution did not significantly differ by income level, national suicide rates, or average exposure levels. Research gaps were found for interactions between air pollution and temperature on suicide risks.

## 1. Introduction

Suicide and suicide attempts (SA) are global concerns with 703,000 deaths/year worldwide and 77% of suicide deaths in low- and middle-income countries in 2019 [1]. Suicide was the eighteenth leading cause of global mortality in 2016 and second leading cause among those age 15–29 years [1,2]. The World Health Organization (WHO) adopted the Mental Health Action Plan 2013–2020 including action goals for improved governance for mental health, mental health and social care services, suicide prevention programs, information system of mental health, and research evidence [3]. 

Suicide involves complicated interactions of various psychological, demographic, biological, and environmental factors [3,4,5,6,7,8,9]. There is an emerging field of research for the complexity of the suicide phenomenon from the intersections between biological and nonbiological factors. For example, neurological diseases can present with psychiatric diseases and intersect for the morphological and functional lesions in brain, which complicates biological models for suicide risk [10,11]. Potential environmental factors affecting suicide include air pollution and ambient temperature [8]. Although not fully identified, a biological rationale for impacts of air pollution and high temperature on suicide, including gene expression extending to the brain, neurotransmitters, and behavioral responses [12], is available from animal studies (Figure 1). Exposure to traffic-related air pollution may cause defects in neural maintenance/regeneration and structural abnormalities in the brain through decreases in brain-derived neurotrophic factor [12]. Particulate matter (PM) and ozone (O_3_) were reported to activate hypothalamic-pituitary-adrenal stress response axis and overactivation or dysfunction of the hypothalamic-pituitary-adrenal axis can lead to depressive behaviors accompanied by elevated plasma levels of adrenocorticotropic hormone and corticosterone [12,13]. High air pollution levels are associated with inflammatory effects on the brain involved in the pathogenesis of neurotoxicity and SA [14]. Diesel exhaust particles and O_3_ may decrease serotonin levels leading to aggressive and depressive behaviors [13]. High temperature can cause acute failure of thermoregulation and, consequently, hyperthermia [15]. High temperature may increase feelings of hostility and aggressive thoughts through imbalance of serotonin and neuroinflammation in the brain, both caused by hyperthermia [16]. High temperature might be associated with higher levels of aggression [17], which is associated with elevated suicidal behaviors [18]. The experience of sudden warmth in the daytime after cold nights can trigger temperature-related overreaction in brown adipose tissue, intensify anxiety, leading to suicidal behaviors [19].

Air pollution and temperature can be highly correlated [20] and are simultaneously exacerbated by climate change (Figure 1) [21], although the correlation differs by pollutant. Air pollution may be a confounding factor for impacts of temperature on suicide and vice versa as both are associated with neuroinflammation in the brain and serotonin neurotransmission. Although possible mechanisms of interaction are not fully understood, increased air pollution during warm periods through open windows and outdoor activities or changes in PM chemical composition are possible pathways [22,23]. Human body thermoregulatory system triggered by heat stress can activate cardiovascular and respiratory reactions, increasing total intake of air pollutants [24]. Temperature can influence transportation and chemical transformation of air pollutants and, therefore, pollutants’ chemical composition and toxicity [25]. Further investigations are needed on interactive effects and confounding between air pollution and temperature. 

Numerous population-based observational studies examined acute impacts of temperature and air pollution on suicide mortality, attempts, and ideation. To date, there are few systematic review studies on these associations [8,19,26,27,28,29,30]. For example, studies reported significant associations between suicide and short-term PM exposure with aerodynamic diameter ≤10 mm (PM_10_) by combining evidence from four studies [29], and PM_10_ and PM with aerodynamic diameter ≤2.5 mm (PM_2.5_) by combining 10 studies [26]. A systematic review of O_3_ and suicide reported that quality of evidence is too low to conclude causal associations [28]. A recent narrative review suggested impacts of PM_10_, PM_2.5_, sulfur dioxide (SO_2_), nitrogen dioxide (NO_2_), O_3_, and carbon monoxide (CO) on suicide [30]. Systematic reviews with meta-analysis for temperature suggested higher suicide rates and emergency room visits for SA associated with high temperature [19,31]. 

The existing findings on air pollution, temperature, and suicide differ among studies. Little is known about how associations differ by study characteristics (e.g., income, climate, air quality, baseline suicide rates). Further, existing studies lack assessments of risk of bias [31], which examines the variation of potential bias to help judge whether the design and conduct of the study compromised believability of the link between exposure and outcome [32,33]. Appraising published studies is essential to aid decision-making in clinical practices and interventions [34]. 

This systematic review and meta-analysis aimed to summarize (1) the influence of air pollution and temperature on suicide; (2) differences in associations between air pollution or temperature and suicide by regional characteristics; and (3) limitations of identified studies (risk of bias) regarding study design and methodologies. Studies of air pollution and temperature vary in lag time (e.g., from daily to yearly scales) [8]. We focused on short-term exposure of days to weeks. This review is unique in its assessment of risk of bias of epidemiological studies as well as its focus on both air pollution and temperature, including potential interactions, in relation to suicide. 

## 2. Materials and Methods 

### 2.1. Literature Search 

The authors performed systematic literature searches using Pubmed, Scopus, and Web of Science for English-language publications before August 2020. Search terms included “air pollution”, “air pollutant”, “PM_2.5_”, “PM_10_”, “particulate”, “temperature”, “ambient temperature”, “air temperature”, “climate”, “weather”, “observation *”, “cohort”, “case control”, “epidemiolog*”, and “suicide”. Truncation filters (*) were used to represent any combination of letters. We performed multiple searches with different combinations of search terms for each database (Appendix A). Grey literature was not searched. Additional studies were identified using Google Scholar from backward reference searching (examining references cited in the included studies) and forward citation chaining (examining papers that cited the included studies) conducted in May 2021.

### 2.2. Study Examination

This research is reported in accordance with the MOOSE Checklist for Meta-analyses of Observational Studies (Appendix A) and the PRISMA statement [35,36]. The authors independently screened titles, abstracts, and full texts of the references in duplication based on our inclusion/exclusion criteria (Table 1). We included population-based observational studies addressing relationships of suicide with daily or weekly changes in ambient air pollutants or temperature. We selected articles with cohort, case-control, time-series, case-crossover, or cross-sectional study designs. Our targeted various outcomes of suicide including complete suicide, suicide attempt (SA), suicidal ideation, self-inflicted injury, and self-harm, and specific International Classification of Diseases (ICD) codes were not used as criteria to include or exclude suicide outcomes in this review. Studies examining exposure to tobacco smoke, occupational exposure to air pollution, or CO poisoning from vehicles were excluded. Observational studies using simplistic statistical methods (e.g., basic correlations) were excluded.

### 2.3. Data Extraction and Synthesis

Using a pregenerated data extraction form, authors independently extracted information of each included study in duplication: author, publication year, study location, study duration, health outcomes (e.g., mortality, hospitalizations), ICD codes, study design, statistical methods, increment of pollution or temperature for presentation of estimated associations, risk estimates (e.g., RRs) and 95% confidence intervals (CIs) or other presentation of study results, considered confounders, and findings for interactions between temperature and air pollution. For air pollution and temperature, we extracted exposure variables (i.e., which variables were considered), exposure methods, period of exposure, and lag period. 

To qualitatively summarize findings, we classified associations into 4 groups: statistically significantly positive, positive but not significant, negative but not significant, and significantly negative.

To quantitatively pool risk estimates, we used random-effect meta-analysis for time-series analysis and time-stratified case-crossover studies examining suicide risk in associations with daily temperature or air pollution, with separate meta-analyses for temperature and each air pollutant. Heterogeneity of included studies was examined by standard *I²* test and publication bias by Egger’s test [37]. When publication bias existed, we used the trim-and-fill method to calculate publication bias-adjusted RR [38]. We included single-city, multi-city, and multi-country studies. For multi-city or multi-country studies, we used pooled estimates in our meta-analysis and when available, used city- or country-specific estimates for analysis of effect modifications.

We conducted random-effect meta-regression analyses for potential effect modifiers. We classified risk estimates by country’s income level using gross national income (GNI) (US$) per capita in 2019 from the World Bank [39] with two groups based on the threshold used by the World Bank: ≥$12,375, <$12,375. The national suicide rates in 2016 from the WHO were matched to locations of included studies and grouped into three tertile groups (<14, 14–15.9, and ≥16 suicides/100,000 population). Study results were categorized into two groups for exposure (separately for temperature and each air pollutant) based on the median of reported average exposure levels across studies. In sensitivity analysis (Appendix A), we applied meta-regression analysis for gross domestic product (GDP) per capita (US$) and purchasing power parity (PPP) per capita (US$) [39].

### 2.4. Assessment of Risk of Bias

We assessed risk of bias for included studies using the OHAT Risk of Bias Rating Tool for Human and Animal Studies [40]. For each of the six OHAT criteria (selection bias, confounding bias, attrition/exclusion bias, detection bias, selective reporting bias, appropriateness of statistical analysis), each study was rated one of 4 scores: 0 (definitely high risk of bias), 1 (probably high risk), 2 (probably low risk), or 3 (definitely low risk). Scores were then averaged across the 6 criteria, to estimate overall probability of risk of bias in each study. Each study’s average score was classified into one of four groups to estimate overall probability of risk of bias: definitely high risk of bias (0–0.9), probably high risk (1–1.9), probably low risk (2–2.7), or definitely low risk (≥2.8). Assessment of risk of bias for each study was conducted by the same two investigators who conducted screening and data extraction for that study. When assessment differed by investigator, the study was revisited by this study’s first author. Overall risk of bias was separately rated for each exposure factor (i.e., temperature, air pollutant) across studies. 

## 3. Results

### 3.1. Characteristics of the Included Studies 

The searches initially identified 2274 unique articles (Figure 2). After excluding ineligible studies and adding additional articles found from citation chaining, 50 studies were eligible for data synthesis. Of these, 18 studies investigated air pollution and 32 investigated temperature. The major reason for exclusion during full-text screening was use of simplistic statistical analyses (*n* = 25).

Study designs varied among the 50 included studies (Table 2). Time-series analysis and case-crossover designs were the most dominant study designs, with 6 (33.3%) time-series and 12 (66.7%) time-stratified case-crossover studies of the 18 air pollution studies, and 17 (53.1%) time-series and 9 (28.1%) time-stratified case-crossover studies of the 32 temperature studies. 

Overall, South Korea (*n* = 9), mainland China and the Taiwan (*n* = 9), US (*n* = 7), and Japan (*n* = 7) were the most the most studied countries. The 18 air pollution studies were based in 9 countries, while the 32 temperature studies were conducted in 26 countries. 

Four (22.2%) of the air pollution studies and 5 (15.6%) of the temperature studies considered hospital visits of SA, whereas the others considered mortality. About 34% (*n* = 17) studies of the total 50 studies did not provide ICD codes. Nine different sets of ICD codes were used to define suicide cases: ICD-10 X60-84 (*n* = 13, 39.4% of the 33 studies that provided ICD codes); ICD-9 E950-E958 and ICD-10 X60-X84 (*n* = 6, 18.2%); ICD-9 E950-E959 (*n* = 4, 12.1%); ICD-9 E950-E958 (*n* = 2, 3.0%); ICD-10 X60-X84 and Y87.0 (*n* = 2, 3.0%); ICD-10 X60-X84 and Y10-Y34 (*n* = 1, 3.0%); ICD-9 E950-E959 and ICD-10 X60-X84 (*n* = 2, 3.0%); ICD-9 E950-E959, E980-E989 (excluding E988.8), ICD-10 X60-84, and Y10-Y34 excluding Y33.9 (*n* = 1, 3.0%); and ICD-9 E950-E959 and E980-E989 (*n* = 1, 3.0%). 

Most air pollution studies (*n* = 14, 77.8%) examined multiple air pollutants: PM_10_ (*n* = 11, 64.7%), PM_2.5_ (*n* = 9, 50%), O_3_ (*n* = 9, 50%), NO_2_ (*n* = 10, 55.6%), SO_2_ (*n* = 9, 50%), and CO (*n* = 5, 27.8%). One study focused on days with Asian dust storms. While the majority of air pollution studies (*n* = 16, 88.9%) used air pollution data from monitors, one used kriging approaches. All air pollution studies considered daily changes in air pollution, except one that considered weekly changes.

Three of the 32 temperature studies adjusted for potential confounding by air pollution, including one of the 18 time-series and case-crossover studies included in our meta-analysis, however, temperature was adjusted in all 13 air pollution studies included in the meta-analysis. In summary, studies adjusted for potential confounding by temperature in assessing suicide risks of air pollution and rarely adjusted for potential confounding by air pollution in estimating associations between temperature and suicide. 

Three studies addressed possible interactions between temperature and air pollution for suicide risks. Thilakaratne et al. [84] estimated hospitalization risks from SA associated with short-term CO and NO_2_ separately for warm and cold seasons, finding higher associations in warm seasons in California, US. Yang et al. [13,86] estimated associations between O_3_ and daily suicide rates separately for warm and cold days (mean temperature > or <23 °C), finding higher associations on cold days in Taipei, Taiwan. 

### 3.2. Risk Summarization 

The 50 studies were classified into three summary groups based on direction and statistical significance of effect estimates (Appendix A). No studies reported significantly negative associations. Studies for temperature generally reported positive associations between suicide and temperature. 

For PM, 35.3% (*n* = 6) of the 11 studies for PM_10_ and six studies for PM_2.5_ suggested positive and significant associations between short-term PM exposure and suicide. For O_3_ 37.5% (*n* = 3) of the eight studies found significantly positive associations with suicide. No significant associations were observed for SO_2_ or NO_2_. Of the five studies for CO, two showed positive and significant associations. The number of studies for CO was limited, hindering assessment of the direction or presence of associations. 

Meta-analysis included 31 eligible studies based on time-series analysis and case-crossover designs, with 18 for temperature and 13 for air pollution (Table 3). Including multiple risk estimates for different regions, 25 results of risk estimation for temperature were combined in meta-analysis. RR of daily suicide rates per IQR increase in daily temperature (7.1 °C) was 1.09 (95% CI: 1.06, 1.13). Heterogeneity among studies for the temperature-suicide association was high (96.9%). The RR for an IQR PM_10_ increase (7.3 mg/m^3^) was 1.01 (95% CI: 1.00, 1.03) and borderline significant (*p* < 0.10). PM_2.5_ also exhibited a borderline significant RR at 1.02 (95% CI: 1.00, 1.05) for an IQR increase (13.1 μg/m^3^). An IQR (17.7 ppm) increase in daily NO_2_ was significantly associated with increased suicide risks (RR = 1.03, 95% CI: 1.00, 1.07), while O_3_, SO_2_, and CO did not exhibit associations with suicide risks. Substantial heterogeneity in associations was found for PM, O_3_, SO_2_, NO_2_, and CO (*I^2^* 54.1%–71.5%). 

Funnel plots for exposure–outcome combinations are shown in Figure 3. Based on the Egger’s test, there was significant publication bias for PM_10_–suicide associations (*p*-value 0.008 for PM_10_), whereas publication bias was not found for the other air pollutants or temperature. Publication bias-adjusted RR for PM_10_ (RR = 1.01, 95% CI: 1.00, 1.03; *I^2^* = 61.5%) did not change from the RR estimate without such adjustment.

The results of meta-regression analyses for GNI (US$) per capita, national suicide rates, and average exposure levels are shown in Table 4. Only GNI (US$) per capita was a significant effect modifier for the temperature–suicide association. RR of suicide risks associated with an IQR increase in daily temperature in regions with low GNI (US$) per capita (RR = 1.20, 95% CI: 1.14, 1.26) was significantly higher than in regions with higher GNI (US$) per capita (RR = 1.09, 95% CI: 1.07, 1.11). Suicide risks associated with air pollution were not modified by national suicide rates or average air pollution levels. Robust results were found using GDP per capita and PPP per capita (Table 5). 

Summaries of risk of bias are shown in Figure 4. We estimated “probably low risk” of bias for both temperature and air pollution. For air pollution, the risk of bias was relatively higher for detection bias, while more than 80% of the included studies showed definitely low risk of bias for the other five criteria of risk of bias. We concluded that overall risk of bias was definitely low for 50.0% (*n* = 9 of 18) of the air pollution studies and was probably low in the remaining air pollution studies (*n* = 9) (Table 6). R For temperature, definitely high risk of bias was found for confounding bias (18.8% of included studies), detection bias (3.1%), and appropriateness of statistical analysis (6.3%) (Figure 4). Major reasons of higher risk of bias for temperature were estimating temperature from an insufficient number of monitoring stations (i.e., one monitoring station per city), lack of controlling for seasonality in statistical models, and using data of suicides from only one hospital within a city. About 25.0% (8 among 32) of included temperature studies was classified as probably high risk of bias, 68.8% (*n* = 22) as probably low risk of bias, and 6.3% (*n* = 2) had definitely low risk of bias (Table 6). Based on this assessment of bias, the true effect would lie close to the estimate effect from meta-analysis for the studied air pollutants except PM_10_, for which we observed publication bias. For temperature, the true effect would likely be close to the estimate from meta-analysis.

## 4. Discussion

Several review studies, with a wide focus on mental health, examined risk of suicide associated with air pollution [7,26,28,29,30] and temperature [19,31]. Relatively fewer studies exist for suicide and air pollution compared to other health outcomes associated with air pollution (e.g., cardiovascular and respiratory diseases). Review studies for suicide risks include narrative, qualitative, and quantitative risk synthesis. We simultaneously focused on air pollution and temperature providing comprehensive information on quantitative and qualitative risk synthesis, the methodologies used, and studies examining exposures together as potential confounders or effect modifiers. 

Our meta-analysis found significant associations between short-term temperature exposure and suicide. A previous meta-analysis found similar results, with slightly higher risks in tropical zones and regions with the middle national income (defined by the World Bank) [19]. We found significant associations with suicide for PM_2.5_, PM_10_, SO_2_, and NO_2_. Our findings on PM_10_ and suicide are robust compared to a previous meta-analysis [29] that incorporated fewer studies. To our best knowledge, our meta-analyses is the first to quantitatively summarize evidence for suicide and O_3_, SO_2_, and NO_2_. Relatively low number of studies for O_3_ compared to other air pollutants highlights the need for further studies. Our findings for effect modification imply that a country’s suicide rates, air pollution levels, and climate do not explain heterogeneity across studies. While effect modification by GNI per capita was significant for the temperature–suicide relationships, future research is required to investigate effect modifications for associations between suicide and air pollution. 

Interaction between air pollution and temperature in relation to health is critical for planning preventive measures. Public warning systems and surveillance systems for health outcomes associated with temperature can be reinforced during days with high air pollution. Driving factors of such interactions could vary by region. For example, Asian dust storms can become more frequent from acceleration of desertification by climate change [89]. Severe heatwaves in Mediterranean regions led to droughts and wildfires in 2000s, which elevated O_3_ and PM during summers [90]. Climate models predict an overall drying of the land in the eastern Mediterranean and Middle East, which will influence air quality and health [91]. Many studies examined effect modification by season for associations between air pollution and mortality finding consistently higher risks in warmer seasons in North America and Europe [22]. Fewer studies used temperature as an effect modifier for mortality effects of air pollution [92,93]. For instance, a multicountry study in northeast Asia found that associations between several air pollutants and mortality (all cause, cardiovascular, respiratory) increased on days with high temperature, as did temperature-mortality associations on days with high air pollution [94]. Studies considered effect modifications by air pollution for temperature-health associations to a lesser extent [22]. We found only three studies addressing effect modifications by temperature for suicide risks of air pollution; these studies were conducted in California, USA and Taipei, Taiwan [13,84,86]. No included studies examined effect modification by air pollution on temperature-suicide relationships. 

In terms of climate change, fossil-fuel-related emissions simultaneously elevate air pollution and temperature. Hyperthermia caused by failure of thermoregulation can increase intake of air pollutants, which can activate pathways related to SA through neurophysiological and stress response pathways. All 18 air pollution studies in our review adjusted for potential confounding by temperature and most studies adjusted for other meteorological factors (e.g., sunshine duration, relative humidity). However, confounding effects of air pollution were rarely controlled in temperature–suicide studies. Thus, assessment for risk of bias showed higher risk of confounding bias for temperature than air pollution in relation to suicide risks. Some studies argued that confounding by air pollution was small and it should not be considered for confounding of temperature–health associations [95,96]. However, the complex mechanisms of suicide risk suggest that air pollution and high temperature are both associated with neuroinflammation and central nervous system pathways (i.e., serotonin functions). Future temperature–suicide studies should consider potential confounding of air pollution and provide a clear rationale for air pollution’s role in this association. 

Some studies were excluded from our meta-analysis due to different metrics for air pollution and temperature. This includes studies that examined temperature–suicide in associations stratified by quantile of PM_2.5_ [74]; calculated differences of suicide rates between the 95th percentile of temperature and the minimum mortality temperature, which differed by city [71]; and used different temperature metrics such as heatwaves [45,56], diurnal temperature range [55], and perceived temperature [46]. Despite different approaches for exposure measurements, these studies generally supported the conclusion for associations between temperature and suicide risks. 

Various terms for suicide outcomes have been used (e.g., suicide, suicidal behavior, SA, suicidal ideation, self-harm, self-injury, intentional deaths). Different sets of ICD codes were used to determine cases of specific outcomes (e.g., respiratory diseases) in epidemiological studies [97]. For this study, we found nine combinations of ICD codes used to identify suicide outcomes. In addition, other definitions of SA such as operational definitions or the lethality of SA have been used in recent studies (not meeting our inclusion criteria) to examine suicide risk [98]. The impact of these varying definitions lead on heterogeneity of risk estimations warrants future study. 

Our findings suggest that more studies are needed on associations between air pollution and suicide, especially for O_3_ and CO. Strengths of this study is that we summarized evidence both quantitatively and qualitatively, assessed risk of bias and provided sub-group analysis of meta-analysis to identify vulnerable populations. 

With respect to limitations, we did not investigate the suicide risks by sex/gender subgroups. Men and women in different age groups may experience different suicide risks from air pollution. Many studies examined neuropsychological, neuroanatomical, and neurophysiological differences arising from interplay of sex hormones [99] such that ovarian steroids (e.g., estradiol, progesterone) affect the density of certain serotonin receptor sites [100]. Suicide risks may differ by income level of country as age-standardized suicide rates were three times higher in males than females in high-income countries; the male–female ratio for suicide rates was more equal in low- and middle-income countries [1]. Health disparities by sex/gender across countries should be studied further. Risks combined in this meta-analysis were based on varying lag structures. A previous meta-analysis [27] suggested that suicide risks differ between lag days and accumulate over the lag days. Future meta-analysis should be able to summarize evidence for the patterns of suicide risks over lag days. Our findings are based on results for a limited number of countries. In particular, future studies are needed for middle- and low-income southeast Asian countries in tropical zones as suicide risks associated with temperature are higher in tropical zones [19]. 

## 5. Conclusions

This systematic review and meta-analysis comprehensively summarized evidence for suicide risks associated with ambient temperature and air pollution. Significant and positive associations between daily high temperature and suicide were found from studies at probably low risk of bias. Results for associations between air pollution and suicide were based on studies that were determined to have definitely low risk of bias; there was suggestive evidence for associations between suicide and PM_2.5_, PM_10_, NO_2_, and NO_2_, with weak evidence for O_3_, SO_2_, and CO. Subgroup analysis showed higher suicide-temperature associations in high-income countries. As relatively less research has been conducted for air pollution, there is a need for future studies on pollution–suicide associations to better determine quality of evidence and effect modifications. Future studies are needed to verify confounding and interactions between air pollution and temperature in terms of the mechanisms of suicide behaviors. Our review also implies the importance of guidance for air quality and hot weather in clinical suicide prevention efforts.

## Figures and Tables

**Figure 1 ijerph-18-07699-f001:**
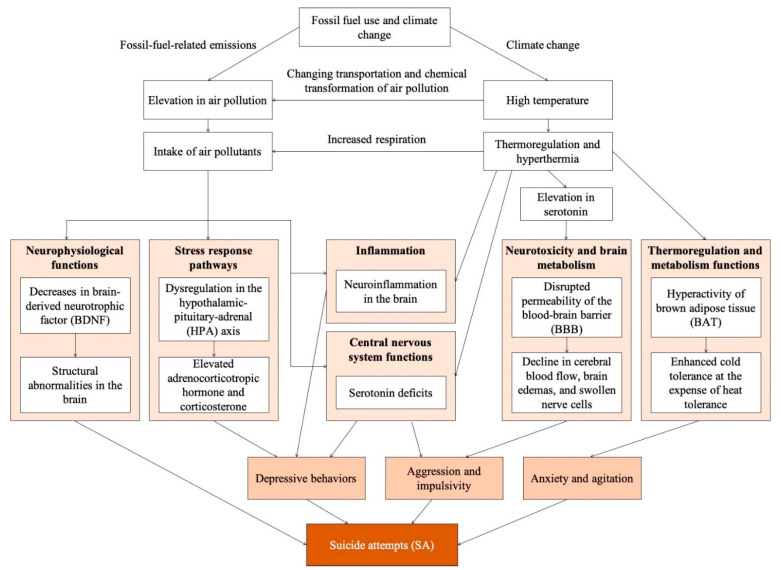
Main putative pathogenic mechanisms of exposure and reciprocal relationships for air pollution, high temperature, and suicide risk.

**Figure 2 ijerph-18-07699-f002:**
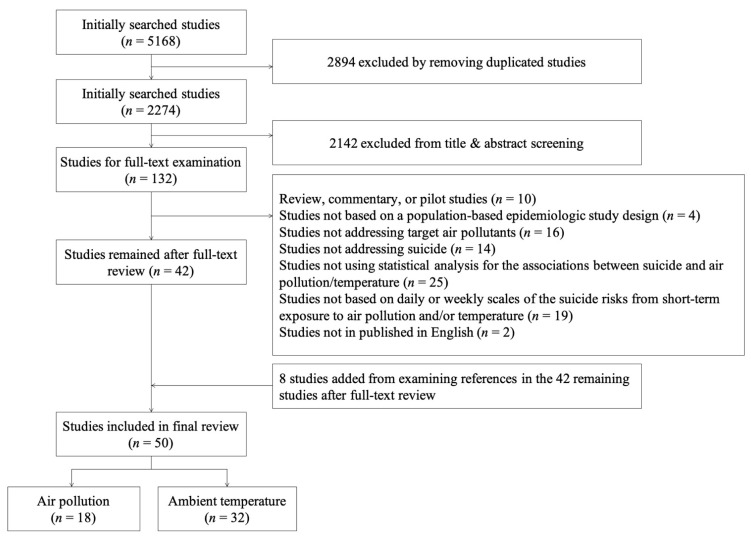
Flowchart of identified studies for systematic review.

**Figure 3 ijerph-18-07699-f003:**
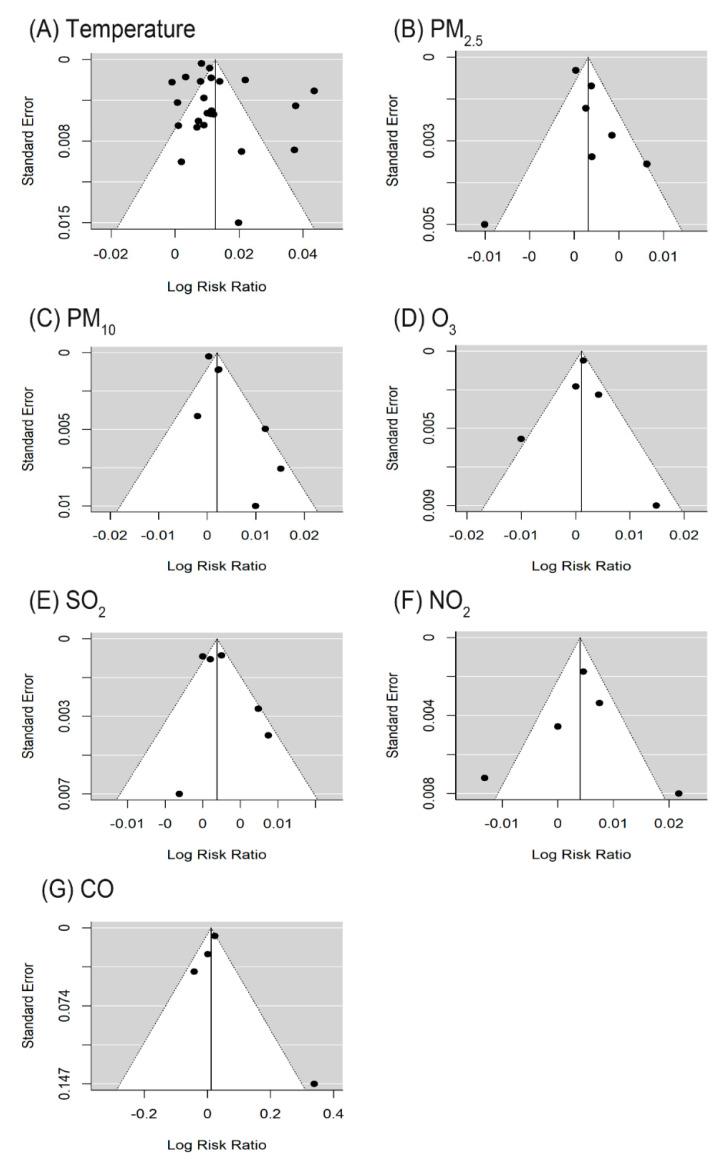
Funnel plots of the meta-analysis for exposure factors (**A**) Temperature; (**B**) PM_2.5_; (**C**) PM_10_; (**D**) O_3_; (**E**) SO_2_; (**F**) NO_2_; (**G**) CO.

**Figure 4 ijerph-18-07699-f004:**
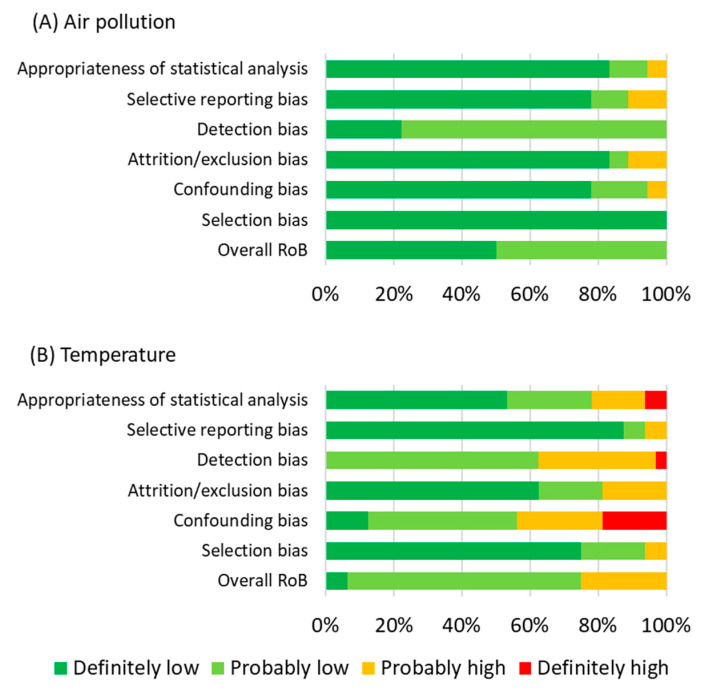
Percent of the included studies by overall risk of bias (RoB) and levels of risk of bias for six criteria: (**A**) air pollution, (**B**) temperature.

**Table 1 ijerph-18-07699-t001:** PICOS inclusion/exclusion criteria.

Parameter	Inclusion	Exclusion
Patient	Health outcomes related with suicide as measured in studies.	N/A
Intervention (exposure)	Air pollutants Ambient temperature	Exposure to tobacco smoke or second-hand smoke. Exposure to occupational air pollution or temperature (high or cold). Exposure to carbon monoxide or benzene poisoning from vehicles.
Comparison	All subpopulations based on biological factors (e.g., age, sex), socioeconomic factors, or contextual factors	N/A
Outcome	Complete suicide, suicide attempt, suicidal ideation, self-inflicted injury, self-harm. Data can be based on mortality data, hospital data, survey data, and etc.	Suicide by carbon monoxide from automobiles (carbon monoxide poisoning).
Study types	Epidemiologic study: population-based observational study such as cohort study, case-control study, cross-sectional study, case-crossover study, survival analysis, time-series analysis, etc.	Systematic review and/or meta-analysis; commentary; editorial articles; news; book or book chapter; meeting report. Research addressing burden of disease rather than exposure-response relationships. Scientific articles not written in English. Not a population-based epidemiological study such as experimental studies, animal studies, studies of toxicology or ecology, etc. Human exposure studies without exposure-response relationship analysis or case report.

**Table 2 ijerph-18-07699-t002:** Characteristics of the included studies.

First Author, Publication Year [Reference]	Location	Study Timeframe	Suicide Deaths: Number or Rate (100,000 Population)	Age Range (Years)	Study Design	Exposure Variables for Air Pollution and/or Temperature	Exposure Methods	Health Outcome	ICD Codes (If Applicable)	Increment of Exposure for Estimates of the Association	Considered Confounders
Astudillo-Garcia, 2019 [41]	Mexico City	2000–2016	Rate: 1.98	≥10	Time-series	PM_10_, PM_2.5_, O_3_, SO_2_, NO_2_	Monitors	Mortality	NR	1 unit	Daily averages for temperature and relative humidity, holiday
Bakian, 2015 [42]	Salt Lake County, Utah, US	2000–2010	1546	All	Case-crossover	NO_2_, PM_2.5_, PM_10_, SO_2_	Monitors	Mortality	NR	IQR increase	Average daily sunlight during the previous 3 days, daily mean temperature, daily mean temperature for the previous 3 days, mean dew point temperature, mean dew point temperature for the previous 3 days, daily mean air pressure, and daily mean air pressure for the previous 3 days.
Bando, 2017 [43]	Sao Paulo, Brazil	1996–2011	Rate: 8.7	NR	Time-series	Weekly minimum temperature	Monitors	Mortality	ICD-10: X60-X84	1 unit	Calendar time for week, weekly mean, minimum, and maximum temperature, insolation hour, irradiation, relative humidity, atmospheric pressure, rainfall
Barker, 1994 [44]	Oxford, England	1976–1989	12,379	NR	Time-series	temperature (monthly)	NR	Suicide cases	NR	1 unit	Month, rainfall, sunshine duration, visibility, cloud cover, wind speed
Basagana, 2011 [45]	Catalonia, Spain	1983–2006	503,389	All	Case-crossover	Temperature	Monitors	Mortality	ICD-10: X60-X84, ICD-9: E950-E959	Dichotomous variable of hot days (95 percentile <) and nonhot days	Day of the week, month
Basu, 2018 [46]	California, US	2005–2013	322,478	All	Time-series	Apparent temperature	Monitors	Hospital (ER) admissions	ICD-9: E950-959	10°F (4.5 °C)	Holidays, day of the week, seasonal trends (calendar time)
Casas, 2017 [47]	Belgium	2002–2011	20,533	≥5	Case-crossover	Daily average PM_10_ and daily maximum 8-h average O_3_	Kriging model	Mortality	ICD-10: X60-84, Y10-34	10 unit increase (microgram/cubic meter)	Day of the week, long-term seasonality, temperature, duration of sunshine
Deisenhammer, 2003 [48]	Tyrol, Austria	1995–2000	702 (518 males, 184 females)	NR	Case-crossover	Temperature	Monitors	Mortality	NR	10 degree of Celsius increase	Mean relative humidity, thunderstorm, rainfall, sultriness, mean atmospheric pressure (tested, but not included in the analytic model)
Dixon, 2018 [49]	9 counties in US	1975–2010	NR	All	Time-series	Daily maximum and minimum temperature	One monitoring station for each county	Mortality	ICD-9: E950-959, ICD-10: X60-X84	NA	Year, week, day of the week
Fernandez-Nino, 2018a [50]	4 cities in Colombia	2011–2014	1942	All	Time-series	PM_2.5_, PM_10,_ O_3_, NO_2_, SO_2_, CO	Monitors	Mortality	ICD-10: X60-X84, Y87.0	increase in 20% of the average exposure level	Temperature, precipitation, relative humidity and holidays
Fernandez-Nino, 2018b [51]	5 cities in Colombia	2005–2015	Rate: 0.06 to 0.57 for men and from 0.01 to 0.14 for women	All	Time-series	Daily mean temperature	One monitoring station	Mortality	ICD-10: X60-X84 or Y87.0	1 unit	Rainfall, holidays, day of the week, month, year
Fountoulakis, 2016 [52]	29 European countries	2000–2012	Rate: 3.8 to 49.0 for men, 0.7 to 14 for women	All	Cross-sectional	Temperature	Modeling data of gridded dataset for daily climate over Europe (termed E-OBS) with a spatial resolution of 0.22 degree	Standardized mortality rate	NR	NR	Economic variables, weather variables.
Grjibovski AM, 2013 [53]	Astana, Kazakhstan	2005–2010	685	All	Time-series	Daily mean temperature, maximum temp, mean apparent temperature, maximum apparent temperature	Monitors	Mortality	ICD-10: X60-X84	1 unit	Month, year, holiday
Hiltunen, 2012 [54]	Helsinki, Finland	1989–1990, 1997–1998	3945	All	Time-series	Daily mean temperature	Monitors	Mortality	NR	1 unit	Global solar radiation, precipitation
Hiltunen, 2014 [55]	Finland	1974–2010	10,802	All	Time-series	Diurnal temperature	Monitors	Mortality	NR	1 unit	21 meteorological factors.
Hu, 2020 [56]	Shenzhen, China	2013–2017	6642 dispatches for suicides	All	Time-series	Daily mean temperature	NR	Ambulance dispatches	ICD-10: X60-X84	Heatwave days: 2 to 4 consecutive days with mean temperature above the thresholds (75, 90, 95, 99th percentiles) versus nonheatwave days	Calendar time, O_3_, CO, NO_2_, PM_10_, PM_2.5_, temperature (same day), holiday, day of the week
Kayipmaz, 2020 [57]	Ankara, Turkey	2017–2019	6777	All	Time-series	Daily mean, maximum, minimum temperature	One monitoring station	Hospital admissions	NR	1 unit	Seasonality, average humidity, average actual pressure (hPa)
Kim, 2010 [58]	7 Metropolitan cities in South Korea	2004	4341	All	Case-crossover	PM_10_, PM_2.5_	Monitors	Mortality	NR	IQR increase	Holidays, mean hours of sunlight from the previous 2 days, temperature, dew point temperature, air pressure
Kim, 2011 [59]	South Korea	2001–2005	49,451	All	Time-series	Daily mean temperature	Monitors	Mortality	ICD-10: X60-X84	1 unit	Sunshine, relative humidity, holidays, long-term trends
Kim, 2015 [60]	16 regions in South Korea	2006–2011	NR	All	Time-series (weekly)	PM_10_, O_3_, SO_2_, NO_2_, CO	Monitors	Mortality	ICD-10: X60-X84	1 unit	Celebrity suicide, average national monthly suicide number for the past 5 years by month matching each weekly (seasonality adjustment), sunlight hours, temperature, consumer price index, unemployment rate, stock index, end-of-week, holiday
Kim, 2016 [61]	15 major cities in Korea, Japan, Taiwan	1972–2010	66024 in Korea, 126,705 in Japan, 17,879 in Taiwan	≥0	Case-crossover	Daily mean temperature	NR	Daily mortality rate	ICD-9: E950.0.E958.9, ICD-10: X60.X84	SD/2-unit (standard deviation of the mean temperature divided by 2) increase	Age, sex, sunshine duration, relative humidity, atmospheric pressure, time trend, month
Kim, 2018 [14]	10 large cities in three Northeast Asian countries (South Korea, Japan, Taiwan)	1979–2010	134,811	All	Case-crossover	PM_10_, PM_2.5_, NO_2_, SO_2_, Coarse PM (PM_2.5–10_)	Monitors	Mortality	ICD-9: E950.0 -E958.9, ICD-10: X60-84	IQR increase	Day of the week, month, year, temperature, sunshine hours, public holidays, relative humidity, sea-level atmospheric pressure, total precipitation
Kim, 2019 [62]	341 locations in 12 countries (Brazil, Canada, Japan, South Korea, Philippines, South Africa, Spain, Switzerland, Taiwan, UK, US, Vietnam)	4–22 years	1320,148	All	Case-crossover	Daily mean temperature (°C)	Monitors	Mortality	ICD-8 and 9: E950.0-E958.9, ICD-10: X60-X84	Between minimum suicide temperature and maximum suicide temperature	Year, month, day of the week, relative humidity (%), the daily total of sunshine duration (hours)
Kubo, 2021 [63]	46 Prefectures, Japan	2012–2015	151,801	≥10	Case-crossover	Daily mean temperature (°C)	Single monitoring station in each city	Emergency room visits	NR	NR	Daily daylight hours, relative humidity
Lee, 2018 [64]	South Korea	2002–2013	73,445	All	Case-crossover	PM_10_, NO_2_, SO_2_, O_3_, CO	Monitors	Mortality	ICD-10: X60-X84	1 unit	Temperature, relative humidity, air pressure, influenza epidemics, holidays
Lee, 2019 [65]	Seoul, South Korea	2002–2015	30,704	All	Case-crossover	Asian dust storms (ADSs)	Modeling data	Mortality	ICD-10: X60-X84	Dichotomous variable for days with and without Asian dust storm	Date of the week, month, year, holiday, temperature, rainfall, sunlight hours, influenza epidemics
Lee, 2020 [66]	Seoul, South Korea	2008–2016	Average daily number: 8	All	Case-crossover	Daily mean temperature (°C)	Monitors	Emergency room visits	Self-harm among ICD-10 S00-T98	24.4 °C: difference between the maximum and minimum injury temperature	Holidays and 2-day moving averages (lag 0–1) of relative humidity, sunlight, and precipitation.
Li, 2018 [67]	Beijing, China	2009–2012	2172	All	Case-crossover	PM_2.5_	Monitors	Mortality	ICD-10: X60-X84	10 unit	Long-term trends, seasonality, year, month, day of the week, temperature, relative humidity
Likhvar, 2011 [68]	47 prefectures, Japan	1972–1995	501,950	All	Time-series	Maximum temperature on the same day	Monitors	Daily suicide death	ICD-9: E950-E958, ICD-10: X60-X84 (Non-violent: E950.0-E952.9; X60-X69, violent E953.0-E958.9; X70-X84)	1 unit	Time, humidity, pressure, sunshine duration, holiday, day of the week
Lin, 2016 [69]	Guangzhou, China	2003–2012	1550	All	Case-crossover	Daily PM_10_, SO_2_, NO_2_	Monitors	Mortality	ICD-10: X64-X84 (non-violent (X60-69), violent (X70-84))	IQR increase	Month, year, day of the week, weather factors (mean temperature, relative humidity, atmospheric pressure, sunshine duration)
Liu, 2019 [70]	Beijing, China	2008–2014	61,384	All	Time-series	PM_2.5_	Monitors	Emergency ambulance dispatch	NR	10 μg/m^3^ increase	Day of the week, public holiday, temperature, relative humidity, calendar time, sunlight duration
Luan, 2019 [71]	31 Metropolitan cities in China	2008–2013	39,347	All	Time-series	Temperature	Monitors	Mortality	ICD-10: X60-X84	Differences between the 95th percentile of temperature and the minimum mortality temperature (MMT)	Humidity, day of the week, holiday, calendar date
Makris, 2021 [72]	Sweden	2006–2012	Total cases 1981	All	Nested case-control	Temperature	Monitors	Hospital visits	ICD-10: X60-X84	1 unit	Previous suicide attempt, county, sex, age, year, and season when the antidepressant treatment was initiated
Maes, 1994 [73]	Belgium	1979–1987	NR	All	Cross-sectional	Weekly average temperature	One monitoring station in each region	Mortality	ICD-9: E950-E958 (violent: E953-E957, nonviolent: E950-E952 & E958)	1 unit	Relative humidity, air pressure, hours of sunlight and precipitation per day, wind speed, geomagnetic index
Merrill, 2019 [74]	US	2011–2015	182,140	All	Cross-sectional	Average maximum daily temperature, average PM_2.5_	Monitors	Mortality rate	NR	NA	Age, sex, race, average daily sunlight, altitude, average PM_2.5_, average daily precipitation, percent living in poverty, percent of adults who smoke cigarettes, percent urban residents, percent obese, percent leisure-time physical inactivity.
Muller, 2011 [75]	Mitte franken, Germany	1998–2005	2987	All	Cross-sectional ecologic study	Temperature, radiation	Monitors	Mortality	NR	1 unit	Season
Ng, 2016 [76]	Tokyo, Japan	2001–2011	29,939	All	Case-crossover	PM_2.5_, suspended particulate matter (SPM), SO_2_, NO_2_	Monitors	Mortality	ICD-10: X60-X84	IQR increase	Holidays, new year holiday season, temperature, relative humidity
Nguyen, 2021 [77]	California, US	2005–2013	193,530	All	Time-series	O_3_, PM_2.5_	Nearest monitor	Emergency room visits	ICD-9: E950-959	10 ppb increase in O_3_ and 10 µg/m^3^ increase in PM_2.5_	NO_2_, apparent temperature, seasonal/long-term trends, public holiday
Page, 2007 [78]	England and Wales	1993–2003	53,623	All	Time-series	Daily mean temperature, suspended high temperature days	Monitors	Daily suicide counts	ICD-9: E950.0-E959.0, E980.0-E989.0 excluding E988.8; ICD-10: X60-X84, Y10-Y34 excluding Y33.9	1 unit	Month, day of the week, holidays, daylight duration
Salib, 1997 [79]	North Cheshire, England	1989–1993	197	All	Case-crossover	Maximum temperature, minimum temperature	One monitoring station	Mortality	ICD-9: E950-959, ICD-9: E980-989	1 unit	Total rainfall, sunshine hours, maximum relative humidity
Santurtún, 2020 [80]	Madrid and Lisbon, Spain	2002–2012	Average daily suicide rate: 3.30/100,000 inhabitants in Madrid and of 7.92/100,000 inhabitants in Lisbon	All	Time-series	Apparent Temperature (AT)	Monitors	Mortality	ICD-10: X60-X84	17.4: difference between maximum AT and the median AT	NO_2_, the day of the week, and time
Schneider, 2020 [81]	4 Bavarian cities and 11 Bavarian counties, Germany	1990–2006	10,595	All	Case-crossover	Temperature	Monitors	Mortality rate	ICD-9: E950.E958, ICD-10: X60-X84	1 unit	Long-term time trend, precipitation, cloud cover
Sim, 2020 [82]	47 prefectures, Japan	1972–2015	Average prefecture-specific annual count of suicide: 516.2	≥15	Case-crossover	Temperature	One monitoring station	Mortality	E950.0-E958.9 for ICD-8 and 9; X60-X84 for ICD-10.	21.5: difference between the maximum suicide temperature versus 5th temperature percentile	Calendar year, month, and day-of-week, humidity
Szyszkowicz, 2010 [83]	Vancouver, Canada	1999–2003	1605	All	Case-crossover	NO_2_, SO_2_, O_3_, CO, PM_10_, PM_2.5_	Monitors	Hospital admissions	NR	IQR increase	Temperature, relative humidity, date (day, month, year), day of the week
Thilakaratne, 2020 [84]	California, US	2005–2013	63,108	All	Time-series	NO_2_, CO	Monitors	Hospital admissions	ICD-9: E950-E959	IQR increase	Daily mean apparent temperature, holidays, day of the week, seasonal/long-term trends
Williams, 2016 [85]	New Zealand	1993–2009	47,265	All	Time-series	Temperature	Interpolation	Hospital admissions	ICD-9: E950-E958	1 unit	Season, population
Yang, 2019a [86]	Taipei, Taiwan	2004–2008	2001	All	Case-crossover	O_3_	Monitors	Daily mortality	ICD-9: E950-E959	IQR increase	Month, day of the week, temperature (lag 0), humidity (lag 0), pollutants (lag 0-2). In two-pollutants models, PM_10_, NO_2_, SO_2_, and CO were adjusted
Yang, 2019b [13]	Taipei, Taiwan	2008–2012	4752	All	Case-crossover	O_3_	Monitors	Hospital admissions	NR	IQR increase	Temperature (lag 0), humidity, day of the week
Yarza, 2020 [87]	Southern Israel	2002–2017	2338 (age 16–90 years)	16–90	Case-crossover, time-series	Daily average temperature	One monitoring station	Hospital admissions	NR	5-unit	Day of the week, month, relative humidity, age, sex, ethnicity, psychiatric diagnosis, SES
Zerbini, 2018 [88]	Sao Paulo, Brazil	2006–2007	NR	NR	Cross-sectional	Perceived temperature	One monitoring station	Mortality	NR	1 unit	None

NR = not reported, NA = not applicable, ICD = International Classification of Diseases, IQR = interquartile range, SES = socioeconomic status. Time-series studies are based on daily analysis unless otherwise specified.

**Table 3 ijerph-18-07699-t003:** Results of meta-analysis for the relative risk of suicide for an interquartile range (IQR) increase in temperature, PM_2.5_, PM_10_, O_3_, SO_2_, NO_2_, and CO.

Variable	Number of Included Results	RR	95% CI	*p*-Value	I^2^
Temperature (°C)	25	1.09	(1.06, 1.13)	<0.0001	96.9%
PM_2.5_ (µg/m^3^)	7	1.02	(1.00, 1.05)	0.098	61.4%
PM_10_ (µg/m^3^)	7	1.01	(1.00, 1.03)	0.054	62.3%
O_3_ (ppb)	5	1.02	(0.96, 1.10)	0.491	54.1%
SO_2_ (ppb)	5	1.02	(1.00, 1.04)	0.222	71.5%
NO_2_ (ppb)	6	1.03	(1.00, 1.07)	0.041	64.1%
CO (ppm)	4	1.02	(0.95, 1.08)	0.631	61.5%

IQR increase: 7.1 °C for temperature, 7.3 µg/m^3^ for PM_10_, 13.1 µg/m^3^ for PM_2.5_, 21.3 ppb for O_3_, 3.7 ppb for SO_2_, and 1.4 ppm for CO.

**Table 4 ijerph-18-07699-t004:** Relative risks (RRs) of suicide associated with an interquartile range increase in temperature, PM_2.5_, PM_10_, and O_3_, SO_2_, NO_2_, and CO by GNI per capita, national suicide rates, and average exposure levels.

Exposure and Effect Modifier	*N*	RR (95% CI)	*p*-Value
Temperature			
GNI (US$) per capita			
Higher (≥$12,375)	31	1.09 (1.07, 1.11)	0.001
Lower (<$12,375)	9	1.20 (1.14, 1.26)	
National suicide rates (per 100,000 population)			
High (≥16)	5	1.11 (1.06, 1.16)	0.964
Medium (14–15.9)	12	1.10 (1.07, 1.13)	
Low (<14)	23	1.11 (1.08, 1.14)	
Average exposure level ^a^			
High	17	1.18 (1.13, 1.23)	0.235
Low	18	1.14 (1.10, 1.18)	
PM_2.5_			
GNI (US$) per capita			
Higher ($12,375)	5	1.01 (1.00, 1.03)	0.519
Lower (<$12,375)	4	1.03 (1.00, 1.06)	
National suicide rates (per 100,000 population)			
High (≥16)	2	1.02 (0.97, 1.06)	0.830
Medium (14–15.9)	0	-	
Low (<14)	6	1.02 (0.99, 1.05)	
Average exposure level ^a^			
High	4	1.01 (0.98, 1.05)	0.864
Low	4	1.02 (0.98, 1.05)	
PM_10_			
GNI (US$) per capita			
Higher (≥$12,375)	10	1.01 (0.99, 1.03)	0.386
Lower (<$12,375)	6	1.01 (0.99, 1.03)	
National suicide rates (per 100,000 population)			
High (≥16)	6	1.01 (1.00, 1.01)	0.680
Medium (14–15.9)	3	1.00 (0.99, 1.01)	
Low (<14)	7	1.00 (0.99, 1.01)	
Average exposure level ^a^			
High	7	1.00 (1.00, 1.01)	0.893
Low	8	1.00 (1.00, 1.01)	
O_3_			
GNI (US$) per capita			
Higher (≥$12,375)	2	1.01 (0.91, 1.29)	0.417
Lower (<$12,375)	3	0.99 (0.86, 1.13)	
National suicide rates (per 100,000 population)			
High (≥16)	1	1.37 (0.93, 2.03)	0.138
Medium (14–15.9)	0	-	
Low (<14)	4	1.02 (0.96, 1.08)	0.129
Average exposure level ^a^			
High	2	0.94 (0.83, 1.08)	0.129
Low	3	1.07 (0.97, 1.18)	
SO_2_			
GNI (US$) per capita			
Higher (≥$12,375)	9	1.02 (1.00, 1.04)	0.139
Lower (<$12,375)	4	0.99 (0.96, 1.03)	
National suicide rates (per 100,000 population)			
High (≥16)	5	1.02 (0.99, 1.06)	0.357
Medium (14–15.9)	4	1.02 (1.00, 1.04)	
Low (<14)	4	0.99 (0.96, 1.03)	
Average exposure level ^a^			
High	6	1.03 (0.97, 1.10)	0.908
Low	7	1.04 (0.97, 1.11)	
NO_2_			
GNI (US$) per capita			
Higher (≥$12,375)	10	1.03 (1.01, 1.05)	0.938
Lower (<$12,375)	5	1.03 (0.97, 1.09)	
National suicide rates (per 100,000 population)			
High (≥16)	5	1.03 (0.99, 1.06)	0.871
Medium (14–15.9)	4	1.03 (0.99, 1.06)	
Low (<14)	6	1.04 (0.99, 1.10)	
Average exposure level ^a^			
High	6	1.02 (1.00, 1.05)	0.015
Low	7	1.04 (1.01, 1.07)	
CO			
GNI (US$) per capita			
Higher (≥$12,375)	4	1.02 (0.95, 1.08)	-
Lower (<$12,375)	0	-	
National suicide rates (per 100,000 population)			
High (≥16)	2	1.02 (0.94, 1.11)	0.877
Medium (14–15.9)	0	1.01 (0.87, 1.16)	
Low (<14)	2		
Average exposure level^a^			
High	2	1.02 (0.94, 1.11)	0.877
Low	2	1.01 (0.87, 1.16)	

^a^: Average exposure level for each exposure factor in the study region. The groups (high, low) were classified based on the median of the exposure factor.

**Table 5 ijerph-18-07699-t005:** Relative risks (RRs) and 95% CIs of suicide associated with an interquartile range (IQR) increase in temperature, PM_2.5_, PM_10_, and O_3_ by GDP per capita and PPP per capita (US$).

Exposure	Group of GDP (US$) per Capita					Group of PPP (US$) per Capita				
	Lower (<$30,025)		Higher (≥$30,025)		*p*-Value	Lower (<$37,058)		Higher (≥$37,058)		*p*-Value
	N	RR (95% CI)	N	RR (95% CI)		N	RR (95% CI)	N	RR (95% CI)	
Temperature (°C)	9	1.20 (1.14, 1.26)	31	0.92 (0.87, 0.98)	0.005	9	1.23 (1.15, 1.31)	29	1.11 (1.08, 1.14)	0.003
PM_2.5_ (µg/m^3^)	4	1.01 (0.99, 1.03)	4	1.01 (0.99, 1.03)	0.839	4	1.01 (0.99, 1.03)	4	1.01 (0.99, 1.03)	0.839
PM_10_ (µg/m^3^)	6	1.00 (0.99, 1.01)	10	1.01 (1.00, 1.01)	0.386	6	1.00 (0.99, 1.01)	10	1.01 (0.99, 1.02)	0.386
O_3_ (ppb)	3	0.99 (0.90, 1.10)	1	1.28 (0.92, 1.78)	0.144	3	0.90 (0.90, 1.10)	1	1.28 (0.92, 1.78)	0.144
SO_2_ (ppb)	4	1.00 (0.98, 1.01)	9	1.01 (1.00, 1.02)	0.139	4	1.00 (0.98, 1.01)	9	1.01 (1.00, 1.02)	0.139
NO_2_ (ppb)	5	1.02 (0.98, 1.05)	10	1.02 (1.00, 1.03)	0.938	5	1.02 (0.98, 1.05)	10	1.02 (1.00, 1.03)	0.938
CO (ppm)	0	-		1.02 (0.95, 1.08)	-	0	-		1.02 (0.95, 1.08)	-

IQR increase was 7.1 °C for temperature, 7.3 µg/m^3^ for PM_10_, 13.1 µg/m^3^ for PM_2.5_, 21.3 ppb for O_3_, 3.7 ppb for SO_2_, and 1.4 ppm for CO.

**Table 6 ijerph-18-07699-t006:** Summary of risk of bias for each study.

First Author’s Last Name & Publication Year	Exposure	Selection Bias	Confounding Bias	Attrition/Exclusion Bias	Detection Bias	Selective Reporting Bias	Appropriateness of Statistical Analysis	Final Score of Overall Risk of Bias	Included in Meta-Analysis (Yes, No)
Astudillo-Garcia, 2019	AP	3	3	3	2	3	3	2.8	Yes
Bakian, 2015	AP	3	3	3	2	1	3	2.5	Yes
Casas, 2017	AP	3	3	3	3	2	3	2.8	No
Fernandez-Nino, 2018	AP	3	3	3	2	3	3	2.8	Yes
Kim, 2010	AP	3	3	3	2	3	2	2.7	Yes
Kim, 2015	AP	3	3	3	2	3	1	2.5	Yes
Kim, 2018	AP	3	3	2	3	3	3	2.8	Yes
Lee, 2019	AP	3	3	3	3	3	3	3.0	No
Lee, 2018	AP	3	3	3	2	3	3	2.8	Yes
Li, 2018	AP	3	3	NR	3	3	3	2.7	Yes
Lin, 2016	AP	3	3	3	2	3	3	2.8	Yes
Liu, 2019	AP	3	3	3	2	3	3	2.8	No
Ng, 2016	AP	3	3	3	2	2	3	2.7	Yes
Nguyen, 2021	AP	3	2	NR	2	3	2	2.2	Yes
Szyszkowicz, 2010	AP	3	2	3	2	1	3	2.3	Yes
Thilakaratne, 2020	AP	3	3	3	2	3	3	2.8	No
Yang, 2019a	AP	3	2	3	2	3	3	2.7	Yes
Yang, 2019b	AP	3	1	3	2	3	3	2.5	No
Bando, 2017	TP	3	2	3	2	1	3	2.3	No
Barker, 1994	TP	1	1	2	1	1	2	1.3	Yes
Basagana, 2011	TP	3	1	3	2	3	3	2.5	No
Basu, 2018	TP	3	3	3	2	3	3	2.8	No
Deisenhammer, 2003	TP	3	0	3	2	3	1	2.0	Yes
Dixon, 2018	TP	3	1	NR	1	3	2	1.8	No
Fernandez-Nino, 2018	TP	3	2	2	1	3	3	2.3	Yes
Fountoulakis, 2016	TP	2	1	NR	2	3	2	1.8	No
Grjibovski, 2013	TP	3	3	3	2	3	2	2.7	Yes
Hiltunen, 2014	TP	3	0	3	2	2	2	2.0	No
Hiltunen, 2012	TP	3	1	2	2	3	1	2.0	Yes
Hu, 2020	TP	3	3	2	2	3	3	2.7	No
Kayipmaz, 2020	TP	3	2	3	2	3	3	2.7	Yes
Kim, 2019	TP	3	2	2	2	3	3	2.5	Yes
Kim, 2011	TP	3	2	3	2	3	2	2.5	Yes
Kim, 2016	TP	3	2	NR	2	3	3	2.3	Yes
Kubo, 2021	TP	3	1	3	1	3	3	2.3	No
Lee, 2020	TP	3	2	3	2	3	3	2.7	Yes
Likhvar, 2011	TP	2	2	3	1	3	3	2.3	Yes
Luan, 2019	TP	3	3	3	2	3	3	2.8	No
Maes, 1994	TP	3	1	NR	NR	3	1	1.7	No
Makris 2021	TP	2	0	3	1	3	2	1.8	No
Merrill, 2019	TP	2	2	2	2	3	2	2.2	No
Muller, 2011	TP	3	0	3	2	2	1	1.8	Yes
Page, 2007	TP	3	2	3	1	3	3	2.5	Yes
Salib, 1997	TP	2	0	3	1	3	1	1.7	No
Santurtún, 2020	TP	3	2	NR	2	3	3	2.3	Yes
Schneider, 2020	TP	3	2	3	2	3	3	2.7	Yes
Sim, 2020	TP	3	2	3	1	3	3	2.5	Yes
Williams, 2016	TP	3	1	3	2	3	0	2.0	Yes
Yarza, 2020	TP	1	2	3	1	3	3	2.2	Yes
Zerbini, 2018	TP	2	0	NR	0	3	0	1.0	No

AP: air pollution, TP: temperature, NR: not reported. The score of ‘NR’ was considered as 1 in the OHAT tool. Colors of cells indicate levels of risk of bias (dark green: definitely low risk of bias, light green: probably low risk of bias, yellow: probably high risk of bias, red: definitely high risk of bias).

## Data Availability

Data are contained within the article or Appendix A.

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
