# Peer review of "Suicide and Associations with Air Pollution and Ambient Temperature: A Systematic Review and Meta-Analysis"

_ijerph, 2021, doi:10.3390/ijerph18147699_

Round 1
Reviewer 1 Report
In this article, the authors analyze articles published on PubMed, Scopus, etc.. and try to analyze the interactions between air pollution, pm3, Temperature, and suicide rates.
In my opinion, the articles used in this study are in the right amount and very well chosen.
The article is well written, clear, and simple.
It's a pity that no one before analyzed this argument and thanks to the authors, this possible correlation was highlighted.
I can not find errors in this paper, I only have one suggestion:
In the introduction, line 33, the authors report data from 2016. Are available more recent data? If you can find them, please change them to a more recent article.
Author Response
In this article, the authors analyze articles published on PubMed, Scopus, etc.. and try to analyze the interactions between air pollution, pm2.5, Temperature, and suicide rates.
In my opinion, the articles used in this study are in the right amount and very well chosen.
The article is well written, clear, and simple.
It's a pity that no one before analyzed this argument and thanks to the authors, this possible correlation was highlighted.
I can not find errors in this paper, I only have one suggestion:
In the introduction, line 33, the authors report data from 2016. Are available more recent data? If you can find them, please change them to a more recent article.
A: Thank you for reviewing our manuscript and for the positive comments. We have replaced the mentioned reference (WHO’s report in 2016) with the most recent report for the global suicide trends in 2019 published in 2021 (see lines 3-4).
Reviewer 2 Report
Please, see the attached file.
Thank you very much.
Best regards.

Author Response
A highly topical subject;
- I personally very much enjoyed the distinction in A) Air pollution and B) Temperature. I believe this helps both to untangle the subject, which is not easy to deal with, and to increase the organization and readability of the paper. Moreover, I appreciated the table resuming the 50 included studies.
As for the main specific sections:
- The Introduction incisively introduces the issues that the review will develop next;
- The Methods and Results are thoroughly and appear rigorous; the PRISMA and MOOSE criteria have been met.
- The Discussion is well drafted, coherently with the aim of review.
- The Limitation section is present.
I believe the work could benefit from one minor revision, which the authors are encouraged to address:
A: Thanks for your comments on our manuscript. We have revised our manuscript according to your suggestions.
- Authors may use the acronym SA referring to suicide attempts. It is commonly used in suicidology. It may be useful to lighten the text.
A: We have changed “suicide attempts” to SA throughout the manuscript.
- Introduction section. One sentence (lines 39-40) mentions the multiplicity and complexity of risk factors involved in suicide risk. Some other selected references to very important and well-known works would enrich the cultural value of this part: “Suicide” _Lancet 2009, 373, 1372–1381, https://doi.org/10.1016/S0140-6736(09)60372-X; “Stress, genetics and epigenetic effects on the neurobiology of suicidal behavior and depression” European Psychiatry 2010, 25(5), 268–271. https://doi.org/10.1016/jeurpsy.2010.01.009; “The neurobiology of suicide” _Lancet Psychiatry 2014, 1(1), 63–72, https://doi.org/10.1016/S2215-0366(14)70220-2; “Suicide and suicidal behavior”, Lancet 2016, 387, 1227–1239. https://doi.org/10.1016/S0140-6736(15)00234-2; “Suicide and suicide risk” Nat Rev Dis Primers, 5(1), 74, doi:10.1038/s41572-019-0121-0. These references are also important in relation to the role of epigenetics, which constitutes one of the possible etiopathogenetic mechanisms central to the topic discussed in this paper.
A: Thank you for suggesting these references, which are very helpful regarding the mechanisms we suggested in our manuscript. We have added these 5 references in lines 40-41.
- Introduction section (line 40). Alongside the different types of suicide risk factors rightly mentioned by the authors (see previous commentary), it would be interesting to introduce through just one simple mention or sentence to a currently emerging field of research, namely how these factors (biological and non-biological) intersect and thereby increase the complexity of the suicide phenomenon. I propose the following works, which refer specifically to the example of neurological diseases, but are broadly generalizable: “When sick brain and hopelessness meets: some matters on suicidality in the neurological patient”, CNS Neurol Disord Drug Targets 2020, doi: 10.2174/1871527319666200611130804; “Neurological diseases and suicide: from neurobiology to hopelessness”, Revue Médicale Suisse, 11(461), 402-405, PMID: 25895218.
A: Thanks for suggesting this. We added new sentences in lines 41-44 to mention the complexity of suicide risk and added the 2 references you suggested.
- Figure 1. This figure is very original. In the title of the figure, I would add "putative" before mechanisms. I would add also "and their reciprocal relationships" or a similar phrase.
A: We have changed the title of Figure 1 to “Main putative pathogenic mechanisms of exposure and reciprocal relationships for air pollution, high temperature, and suicide risk.”
- ICD codes: it would be helpful to specify the corresponding diagnoses (barring errors on my part, I have not found a sentence or agenda in which they are specified).
A: Thanks for this comment. We included studies examining various suicide outcomes as noted in each study and did not include/exclude studies based on certain ranges of ICD codes. To better clarify this, we added the following new sentence to lines 126-129: “Our targeted various outcomes of suicide including complete suicide, suicide attempt, suicidal ideation, self-inflicted injury, and self-harm, and specific International Classification of Diseases (ICD) codes were not used as criteria to include or exclude suicide outcomes in this review.” The table summarizing characteristics of the included studies (now Table 2, formerly Table 1) specifies the ICD code used in each study or notes if such codes were not included.
- Again, barring errors on my part, I would specify in the text and in the algorithm the wording PRISMA criteria (it seems to me that they have not been explicitly mentioned).
A: We added new text for PRISMA in line 121 and added a new reference of “Liberati, Alessandro, et al. "The PRISMA statement for reporting systematic reviews and meta-analyses of studies that evaluate health care interventions: explanation and elaboration." Journal of clinical epidemiology 62.10 (2009): e1-e34.”
- Discussion section. It would need to add in the Discussion and debate this very recent work: “Meteorological Variables and Suicidal Behavior: Air Pollution and Apparent Temperature are Associated with High =-lethality Suicide Attempts and Male Gender” Front Psychiatry 2021, 12:653390. doi: 10.3389/fpsyt.2021.653390.
A: Thanks for this comment. We added this reference in lines 364-366 and added text highlighting their approaches to identify suicide outcomes.
- The paper would benefit from a brief explicit mention of the practical implications (e.g., in the Conclusions section), which would be useful to the clinical reader.
A: Thanks for this comment. Please see lines 423-424 where we added new text on the importance of guidance relating to the environment in clinical approaches for suicide prevention.
- Supplementary material. I found interesting and clear tables 3S (inclusion and exclusion criteria, the table can be a good alternative to their inclusion in the primary text), 5S, 6S, and Figure 1S. Consider including them in the main text instead of supplementary material that is often not read?
A: Thanks for this suggestion. Table S3, S5, S6, and Figure 1S have been moved to the main text and the numbers of them have been changed accordingly (now Tables 1, 5, and 6 and Figure 3).
Reviewer 3 Report
In Abstract, "RR of suicide was significantly higher higher-income than lower-income countries" -->"RR of suicide was significantly higher in higher-income than in lower-income countries"
In Introduction, you need to state that temperature is related to aggression (Bushman BJ, Wang MC, Anderson CA. Is the curve relating temperature to aggression linear or curvilinear? Assaults and temperature in Minneapolis reexamined. J Pers Soc Psychol. 2005;89(1):62-66. doi: 10.1037/0022-3514.89.1.62) and aggression in turn is related to suicide (Hill SY, Jones BL, Haas GL. Suicidal ideation and aggression in childhood, genetic variation and young adult depression. J Affect Disord. 2020;276:954-962. doi: 10.1016/j.jad.2020.07.049. Epub 2020 Jul 24).
The rest is fine.
Author Response
In Abstract, "RR of suicide was significantly higher higher-income than lower-income countries" -->"RR of suicide was significantly higher in higher-income than in lower-income countries"
A: Thank you for noticing this. We corrected the sentence as suggested.
In Introduction, you need to state that temperature is related to aggression (Bushman BJ, Wang MC, Anderson CA. Is the curve relating temperature to aggression linear or curvilinear? Assaults and temperature in Minneapolis reexamined. J Pers Soc Psychol. 2005;89(1):62-66. doi: 10.1037/0022-3514.89.1.62) and aggression in turn is related to suicide (Hill SY, Jones BL, Haas GL. Suicidal ideation and aggression in childhood, genetic variation and young adult depression. J Affect Disord. 2020;276:954-962. doi: 10.1016/j.jad.2020.07.049. Epub 2020 Jul 24). The rest is fine.
A: Thank you for this suggestion. We added new sentences to mention that high temperature is associated with aggression, which is associated with suicidal behaviors. The 2 suggested references have been added as well (see lines 62-64).